# ^99m^Tc-Labeled Iron Oxide Nanoparticles as Dual-Modality Contrast Agent: A Preliminary Study from Synthesis to Magnetic Resonance and Gamma-Camera Imaging in Mice Models

**DOI:** 10.3390/nano12152728

**Published:** 2022-08-08

**Authors:** Maria-Argyro Karageorgou, Aristotelis-Nikolaos Rapsomanikis, Marija Mirković, Sanja Vranješ-Ðurić, Efstathios Stiliaris, Penelope Bouziotis, Dimosthenis Stamopoulos

**Affiliations:** 1Institute of Nuclear & Radiological Sciences & Technology, Energy & Safety, National Center for Scientific Research “Demokritos”, 15341 Athens, Greece; 2Department of Physics, National and Kapodistrian University of Athens, 15784 Athens, Greece; 3Laboratory for Radioisotopes, “Vinča” Institute of Nuclear Sciences, University of Belgrade, P.O. Box 522, 11001 Belgrade, Serbia; 4Institute of Nanoscience & Nanotechnology, National Center for Scientific Research “Demokritos”, 15341 Athens, Greece

**Keywords:** Technetium-99m, dual-modality contrast agents, radiolabeling, in vitro stability, biodistribution, gamma-camera imaging, magnetic resonance imaging

## Abstract

The combination of two imaging modalities in a single agent has received increasing attention during the last few years, since its synergistic action guarantees both accurate and timely diagnosis. For this reason, dual-modality contrast agents (DMCAs), such as radiolabeled iron oxide (namely Fe_3_O_4_) nanoparticles, constitute a powerful tool in diagnostic applications. In this respect, here we focus on the synthesis of a potential single photon emission computed tomography/magnetic resonance imaging (SPECT/MRI) DMCA, which consists of Fe_3_O_4_ nanoparticles, surface functionalized with 2,3-dicarboxypropane-1,1-diphosphonic acid (DPD) and radiolabeled with ^99m^Tc, [^99m^Tc]Tc-DPD-Fe_3_O_4_. The in vitro stability results showed that this DMCA is highly stable after 24 h of incubation in phosphate buffer saline (~92.3% intact), while it is adequately stable after 24 h of incubation with human serum (~67.3% intact). Subsequently, [^99m^Tc]Tc-DPD-Fe_3_O_4_ DMCA was evaluated in vivo in mice models through standard biodistribution studies, MR imaging and gamma-camera imaging. All techniques provided consistent results, clearly evidencing noticeable liver uptake. Our work documents that [^99m^Tc]Tc-DPD-Fe_3_O_4_ has all the necessary characteristics to be a potential DMCA.

## 1. Introduction

The growing concern for the early and accurate diagnosis of diseases, such as cancer, has led to the increased use of iron oxide nanoparticles, namely magnetite, Fe_3_O_4_ and/or maghemite, γ-Fe_2_O_3_, in medical applications, due to their remarkable properties, including, biocompatibility [1,2,3], biodegradability [4], easy synthesis [5] and size-dependent magnetic properties [6] compared to other magnetic nanomaterials. Thus, among their various clinical applications [7], Fe_3_O_4_ nanoparticles have been extensively studied as magnetic resonance imaging (MRI) contrast agents (CAs), due to their ability to alter the spin-lattice (T_1_) or spin-spin (T_2_) relaxation times of the adjacent water protons in the tissue where they accumulate, providing hypointense signals that can efficiently unveil the host pathological tissue from the healthy environment.

Even though Fe_3_O_4_ nanoparticles were mostly synthesized for MR imaging, their radiolabeling enables simultaneous imaging with other relevant modalities, namely positron emission tomography (PET) and/or single photon emission computed tomography (SPECT), thus providing a powerful dual-imaging tool. Indeed, a dual modality contrast agent (DMCA), such as radiolabeled Fe_3_O_4_ nanoparticles, combines the advantages of each imaging technique (i.e., the high sensitivity of PET or SPECT with the high spatial resolution for MRI), providing complementary information. For this reason, several papers [8,9,10,11,12,13,14,15,16] have reported the synthesis and in vivo application of radiolabeled Fe_3_O_4_ nanoparticles as potential SPECT/MRI or PET/MRI DMCAs. In some cases, these DMCAs may also be used for drug delivery and magnetic hyperthermia to achieve both diagnosis and therapy of cancer [17,18,19,20,21].

Technetium-99m (^99m^Tc) is the most widely used radionuclide in nuclear medicine, since more than 80% of the currently performed diagnostic scans are realized with administered ^99m^Tc-based radiopharmaceuticals [22]. For example, ^99m^Tc-labeled bisphosphonates, such as ^99m^Tc-methylene-diphosphonate (^99m^Tc-MDP) and ^99m^Tc-hydroxymethane diphosphonate (^99m^Tc-HDP), are widely used in the detection of osteomyelitis [23] or bone metastasis [24], while ^99m^Tc-labeled nanocolloids have been extensively applied for the detection of the sentinel lymph nodes of the prostate [25]. Additionally, the radiopharmaceutical ^99m^Tc-3,3-diphosphono-1,2-propanodicarboxylic acid (^99m^Tc-DPD), known as TECEOS^®^, is a well-known imaging agent used in bone scintigraphy, while it is currently being investigated in cardiac imaging in order to detect cardiac amyloid in patients with known or suspected amyloidosis [26]. The increased interest for ^99m^Tc is due to its advantageous properties. Firstly, its short half-life of 6 h allows the preparation of the respective radiopharmaceuticals and successful conduction of the examination with the minimum radiation dose to patients. Secondly, its photon emission of 140 keV with 89% abundance is optimal for effective imaging with gamma-cameras. Finally, the in situ elution from a ^99^Mo/^99m^Tc generator makes ^99m^Tc easily available in every day clinical practice [22,27].

Accordingly, the DMCA described in this work consists of Fe_3_O_4_ nanoparticles, which are surface functionalized with 2,3-dicarboxypropane-1,1-diphosphonic acid (noted as DPD) and radiolabeled with ^99m^Tc, that is [^99m^Tc]Tc-DPD-Fe_3_O_4_. The substance DPD was used due to its water solubility and biocompatibility. It also endows the bare Fe_3_O_4_ nanoparticles with high colloidal stability [28], a property that is of highly importance in biomedical applications. As a tetradentate ligand, with two phosphonate and two carboxylate groups, it serves as an effective chelator for radiolabeling with different radionuclides (i.e., ^90^Y, ^68^Ga etc.), providing highly stable conjugates [28]. The DMCA was radiolabeled by a direct reduction method using anhydrous SnCl_2_ and was subsequently evaluated both in vitro and in vivo. Referring to the in vitro evaluation, stability studies were assessed up to 24 h after incubation in phosphate buffer saline (PBS, pH 7.4) and human serum, to roughly estimate the potential dissociation of the radionuclide in vivo. The results showed that the DMCA is highly stable after 24 h of incubation in PBS (~92.3% intact), while it is adequately stable after 24 h of incubation in human serum (~67.3% intact). Referring to the in vivo evaluation, standard biodistribution, MR imaging and gamma-camera imaging studies were performed in mice models. The combined in vivo results clearly documented noticeable liver uptake and further clarified the imaging potential of the specific DMCA.

## 2. Materials and Methods

Technetium-99m is a gamma emitting radionuclide that presents serious health threats. Thus, due to the obvious safety issues associated with ^99m^Tc-based radiochemical processes, they are conducted under special radioprotective precautions to reduce the risk of harm. Part of this research was performed in a licensed radiochemical facility, which has all the necessary infrastructure, licenses and expertise to safely conduct experiments with radionuclides.

All reagents and solvents used in the radiolabeling process were commercially available and used as received without further purification and are as follows: tin(II) choride (SnCl_2_), 98%, anhydrous (Acros Organics, Fisher Scientific, Loughborough, UK), hydrochloric acid (HCl) (TraceSelect, for trace analysis >37%, Sigma-Aldrich, St. Louis, MO, USA), acetone (Emplura, Merck, Dermstadt, Germany), sodium citrate (tri-Natriumcitrat-2-hydrat, Riedel-de Haën, Seelze, Germany) and sodium chloride 0.9% (Fresenius Kabi AG, Bad Homburg, Germany). A commercial ^99^Mo/^99m^Tc generator (Mallinckrodt Medical B.V., St. Louis, MO, USA) was used to elute ^99m^Tc, as Na[^99m^Tc]TcO_4_^−^. Radioactivity of the Na[^99m^Tc]TcO_4_^−^ eluate and [^99m^Tc]Tc-DPD-Fe_3_O_4_ DMCA were measured using a dose calibrator (Capintec, Ramsey, NJ, USA). ITLC-SG glass microfiber chromatography paper impregnated with silica gel (Agilent Technologies, Folsom, CA, USA) and a radio-TLC scanner (LabLogic, Sheffield, UK) were used for the determination of radiolabeling yield and of in vitro stability of the DMCA. Human serum was acquired from Sigma-Aldrich (Merck, Darmstadt, Germany), while isoflurane 1000 mg/mL was acquired from Iso-Vet (Piramal Critical Care, Voorschoten, The Netherlands). The heating of the samples for stability studies was performed on a Digital Thermoblock TD-TDC (Falc Instruments s. r. l, Treviglio, Italy). Water was deionized to 18 MΩ cm using an easy-pure water filtration system (Barnstead International, Dubuque, Iowa). A gamma scintillation counter (Cobra II, Canberra, Packard, Schwadorf, Austria) was used for the determination of the radioactivity of the organs expressed in counts per minute (cpm). Dynamic light scattering (DLS) measurements were carried out by means of an AXIOS-150/EX (Triton Hellas, Thessaloniki, Greece) DLS apparatus, equipped with a 30 mW He-Ne laser emitting at 658 nm and an avalanche photodiode detector at an angle of 90°.

All animal experiments were carried out according to European and national regulations and were further approved by the Ethics Committees of NCSR “Demokritos”. The Animal Housing Facility situated at the Radiochemical Studies Laboratory is registered according to the Greek Presidential Decree 56/2013 (registration numbers: EL 25 BIO 022 and EL 25 BIO 021), in accordance with the European Directive 2010/6,3 which is harmonized with national legislation on the protection of animals used for scientific purposes. The animal care and procedures followed were in accordance with institutional guidelines and licenses issued by the Department of Agriculture and Veterinary Service of the Prefecture of Athens (protocol number: 875110/11-11-2020).

For the realization of the in vivo biodistribution study, *n* = 12 normal female Swiss albino mice (6–8 weeks, weight ~34 gr) were used. The animals were acquired from the breeding facilities of the Institute of Biosciences and Applications, NCSR “Demokritos.” The mice were housed under controlled conditions in a ventilated IVC cage system with constant temperature (22 ± 2 °C) and humidity (45–50%), in addition to a 12 h light/dark cycle and free access to food and tap water.

### 2.1. Radiosynthesis of [^99m^Tc]Tc-DPD-Fe_3_O_4_ DMCA

The synthesis of the parent, non-radiolabeled contrast agent (CA), DPD-Fe_3_O_4_, has been reported in a previous study of ours [28].

Radiolabeling of DPD-Fe_3_O_4_ with ^99m^Tc was performed by direct reduction technique using anhydrous SnCl_2_ as the reducing agent, based on previously reported methods [29,30]. Initially, 8–9 mg of SnCl_2_ were dissolved in 250 μL HCl (>37%) and subsequently, deionized water was added to achieve a total volume of 5 mL. For radiolabeling, 200 μL of phosphate buffer saline (PBS) pH 10 was mixed with 20 μL of DPD-Fe_3_O_4_ DMCA (obtained from a stock sample of initial concentration C_CA_ = 8 mg DPD-Fe_3_O_4_/mL). Then, 50 μL of SnCl_2_ solution was added to the mixture. At this point, the pH of the mixture was 2. Thus, another 300 μL of PBS pH 10 was added to the mixture to adjust the pH at 7. After the pH adjustment, 100 μL of Na[^99m^Tc]TcO_4_^−^ (average activity ~54 MBq) was added and left to be gently stirred at room temperature for 1h.

In order to determine the radiolabeling yield of the DMCA, two systems of ascending instant thin-layer (ITLC-SG) chromatography were employed, using acetone and sodium citrate solution (0.1 M) as the mobile phases. Referring to acetone, the DMCA was separated from the free [^99m^Tc]TcO_4_^−^ and remained at the application point, along with the colloidal ^99m^Tc (thus, R_f_ = 0.0 for both the DMCA and the colloidal ^99m^Tc), while the free [^99m^Tc]TcO_4_^−^ ions migrated with the solvent to the front of the strip (R_f_ = 1.0). Referring to the sodium citrate solution, both the DMCA and the free [^99m^Tc]TcO_4_^−^ ions were separated from the colloidal ^99m^Tc and moved with the solvent to the front of the strip (thus, R_f_ = 1.0 for both the DMCA and the free [^99m^Tc]TcO_4_^−^), while the colloidal ^99m^Tc remained at the application point (R_f_ = 0.0). The % of radiolabeling yield (RY) was calculated by means of a radio-TLC scanner.

#### Dynamic Light Scattering

Dynamic light scattering (DLS) measurements were carried out on the resulting DMCA by means of an AXIOS-150/EX (Triton Hellas) DLS apparatus. In a typical DLS measurement, 200 μL of [^99m^Tc]Tc-DPD-Fe_3_O_4_ DMCA was diluted with 100 μL ultra-pure water and measured at T = 20 °C for 5 min. Ten light scattering measurements were obtained from the diluted sample and subsequently analyzed using CONTIN analysis to obtain the resulting hydrodynamic diameter (D*_h_*).

### 2.2. In Vitro Stability Studies of [^99m^Tc]Tc-DPD-Fe_3_O_4_ DMCA

For the realization of the in vitro stability studies, 100 μL of [^99m^Tc]Tc-DPD-Fe_3_O_4_ DMCA was mixed with 900 μL of PBS (pH 7.4) and incubated on a shaker at room temperature, while 50 μL of DMCA was incubated with 450 μL of human serum at 37 °C. Aliquots were taken at four time points after incubation, namely 1, 2, 4 and 24 h and analysed by ascending ITLC-SG chromatography, as described above.

### 2.3. In Vivo Biodistribution Study of [^99m^Tc]Tc-DPD-Fe_3_O_4_ DMCA in Normal Mice

The in vivo biodistribution of [^99m^Tc]Tc-DPD-Fe_3_O_4_ DMCA in *n* = 12 normal mice was evaluated as follows: 100 μL of PBS (pH 7.4) suspension of [^99m^Tc]Tc-DPD-Fe_3_O_4_ DMCA was administered to the mouse via the tail vein (each mouse received 10 μg DMCA/100 μL). At 1, 2, 4 and 24 h post injection (p.i.), the animals were euthanized (*n* = 3 mice/time point) and subsequently, the organs (i.e., heart, liver, spleen, lungs, kidneys, stomach, intestines, pancreas, bones), samples of blood, muscles and urine were collected, weighed and measured in a gamma counter. For the determination of the injected dose in each mouse, the remaining radioactivity in the tail, as well as background counts, were subtracted. In addition, the measurements were auto-corrected for radioactive decay by the counter. The accumulation of [^99m^Tc]Tc-DPD-Fe_3_O_4_ DMCA in each organ at each time point was expressed as the percentage of injected activity per gram of tissue (% IA/g). All the calculations were performed compared to a standard dose, which corresponded to 1/10 of the injected solution.

### 2.4. In Vivo MR Imaging of [^99m^Tc]Tc-DPD-Fe_3_O_4_ in Normal Mice

MR imaging was conducted at the Alphavet Veterinary Diagnostic Imaging Center, Athens, Greece, using a 1.5 T MRI unit (Signa Creator, GE Healthcare). According to our protocol, *n* = 3 normal Swiss mice (6–8 weeks, weight ~34 gr) were used in a typical MRI experiment, which were intravenously injected via the tail vein as follows: mouse (1) was injected with 100 μL (0.0259 mg) [^99m^Tc]Tc-DPD-Fe_3_O_4_ DMCA, leading to a final body concentration of C_DMCA_ = 0.01 mg [^99m^Tc]Tc-DPD-Fe_3_O_4_ DMCA/mL. Mouse (2) was injected with 100 μL (0.23 mg) [^99m^Tc]Tc-DPD-Fe_3_O_4_ DMCA, leading to a final body concentration of C_DMCA_ = 0.1 mg [^99m^Tc]Tc-DPD-Fe_3_O_4_ DMCA/mL. Mouse (3) was injected with 100 μL (0.24 mg) non-radiolabeled, DPD-Fe_3_O_4_ contrast agent (CA), leading to a final body concentration of C_CA_ = 0.1 mg DPD-Fe_3_O_4_/mL. The calculations of the final concentrations were based on the assumption that the blood content is approximately 7% of the total weight of each mouse.

After 6 h p.i., the mice were anesthetized with an intraperitoneal injection of ketamine (75 mg/kg) and xylazine (5 mg/kg) and placed for imaging, where standard T_1_-weighted corona/axial and T_2_-weighted axial MRI data were acquired, with a 1.5 mm step size between the sections. MRI images have also been acquired from *n* = 3 non-injected mice, which served as the reference group to our study. To check the reproducibility of these in vivo MRI experiments, we have conducted at least two independent runs for every distinct concentration of the parent CA, DPD-Fe_3_O_4_ and of the DMCA.

### 2.5. In Vivo Gamma-Camera Imaging of [^99m^Tc]Tc-DPD-Fe_3_O_4_ in Normal Mice

Following the in vivo biodistribution study of [^99m^Tc]Tc-DPD-Fe_3_O_4_ DMCA, in vivo gamma camera imaging was performed in *n* = 4 normal female Swiss mice (6–8 weeks, weight ~34 g) using a sensitive small-field gamma-camera system, with high spatial resolution (0.95 ± 0.05 mm on planar imaging) [31] dedicated to small-animal and other preclinical studies. In particular, each mouse was injected intravenously via the tail vein with 100 μL/0.02 mg/~7.03 MBq of [^99m^Tc]Tc-DPD-Fe_3_O_4_ DMCA and placed for imaging at 1, 2 and 4 h p.i. At each time point, a 10-min gamma-camera imaging was conducted. During the study, each mouse was anesthetized with an intraperitoneal injection of ketamine (75 mg/kg) and xylazine (5 mg/kg), before being placed for imaging.

## 3. Results and Discussion

### 3.1. Radiosynthesis of [^99m^Tc]Tc-DPD-Fe_3_O_4_ DMCA

It has been reported that the Fe_3_O_4_ nanoparticles can form stable conjugates with phosphonate compounds [8,32,33]. Accordingly, in our study, we radiolabeled a DPD surface-coated Fe_3_O_4_ CA with the photon-emitting ^99m^Tc, using the direct radiolabeling technique. To this effect, in order to achieve a stable conjugate during the radiolabeling process, parameters such as the effect of pH, the amount of SnCl_2_ and the time of incubation were studied.

To determine the most appropriate pH value, in order to achieve high radiolabeling yields of the resulting ^99m^Tc-DPD-Fe_3_O_4_ DMCA, different pH values were investigated, namely pH 2, pH 4 and pH 7. The effect of the pH on the radiolabeling yield is shown in Table 1. Our experimental results indicated that at pH 2, the percentage (%) of radiolabeling yield was found to be 87% at 30 min of incubation, which increased up to 94% at 120 min of incubation at room temperature (RT). By increasing the incubation temperature to T = 50 °C, the % of radiolabeling yield was decreased compared to the results obtained at RT. At pH 4, the % of radiolabeling yield remained quite stable at all time points examined (80.2% at 30 min, 85.2% at 60 min and 88% at 120 min of incubation at room temperature). Similarly, the % of radiolabeling yield at T = 50 °C were found to be decreased compared to the ones obtained at RT. At pH 7, the % of radiolabeling yield was found to be high at all time points examined at room temperature (96.3% at 30 min, 96.4% at 60 min and 97.2% at 120 min of incubation).

For pH 2, the highest radiolabeling yield of [^99m^Tc]Tc-DPD-Fe_3_O_4_ DMCA was obtained at 120 min after incubation at RT, (namely, 94%), while for pH 7, the % of radiolabeling yield of [^99m^Tc]Tc-DPD-Fe_3_O_4_ DMCA was found to be high at all examined time points at RT. Thus, in order to check the reproducibility of these measurements (indicated with “^1^” in Table 1), we have conducted at least three independent runs for each one. The results are summarized in Table 2.

As shown, the % of radiolabeling yield of a [^99m^Tc]Tc-DPD-Fe_3_O_4_ DMCA, which was incubated at pH 2 and at RT for 120 min, was found to be 89.5 ± 6.9%. On the contrary, the % of radiolabeling yield of a [^99m^Tc]Tc-DPD-Fe_3_O_4_ DMCA, which was incubated at pH 7 and at RT for 30, 60 and 120 min, was found to be 95.9 ± 0.8%, 95.8 ± 1.3% and 96.1 ± 1.8%, respectively.

Subsequently, the radiolabeling yields of 89.5 ± 6.9% (120 min, pH 2) and 95.8 ± 1.3% (60 min, pH 7) were evaluated for stability in relation to time. Briefly, the samples were left at the work bench and aliquots were sampled at three time points (60, 120, and 240 min) to evaluate the % in vitro bench stability. The results are included in Table 3. As shown, the sample of [^99m^Tc]Tc-DPD-Fe_3_O_4_ DMCA incubated at pH 2 did not remain stable in relation to time, since a decrease in the % of radiolabeling yield was observed. This result indicates that the radionuclide progressively dissociates from the DPD-Fe_3_O_4_ CA. On the other hand, the sample of [^99m^Tc]Tc-DPD-Fe_3_O_4_ DMCA incubated at pH 7 exhibited excellent stability in relation to time of more than 93% intact DMCA at 240 min. Thus, the pH 7 was selected as the optimum for radiolabeling.

The amount of the reducing agent plays a significant role in radiolabeling with ^99m^Tc. Specifically, an excess of SnCl_2_ for the reduction in ^99m^Tc may provide undesirable colloids, along with the formation of the desired [^99m^Tc]Tc-DPD-Fe_3_O_4_ DMCA. Thus, to achieve high radiolabeling efficiency, the quantity of the SnCl_2_ used in our study was kept within the range of 7.5–9.7 mg. The effect of the reducing agent on the radiolabeling yield of various samples of [^99m^Tc]Tc-DPD-Fe_3_O_4_ DMCA, incubated at pH 7 for 60 min at RT, are shown in Table 4.

Finally, the effect of incubation time on the radiolabeling yield is shown in Table 2. The samples of [^99m^Tc]Tc-DPD-Fe_3_O_4_ DMCA were incubated at pH 7 and RT and indicated high radiolabeling yields at all examined time points after incubation.

An example of ascending ITLC-SG analysis for the determination of the radiolabeling yield of [^99m^Tc]Tc-DPD-Fe_3_O_4_ DMCA incubated for 60 min at room temperature, using acetone (panel a) and sodium citrate solution (0.1 M) (panel b) as the mobile phases, is indicated in Figure 1. Referring to the radio-chromatogram of Figure 1a, both the [^99m^Tc]Tc-DPD-Fe_3_O_4_ DMCA and the colloidal ^99m^Tc remained at the application point of the strip (98.6% of the total radioactivity counts), while the unbound [^99m^Tc]TcO_4_^−^ ions migrated with acetone to the front of the strip (1.34% of the total radioactivity counts). Referring to the radio-chromatogram of Figure 1b, colloidal ^99m^Tc remained at the application point (2.6% of the total radioactivity counts), while both the DMCA and free [^99m^Tc]TcO_4_^−^ ions migrated with the solvent to the front of the strip (97.5% of the total radioactivity counts). From the combination of these two systems, we estimated the radiolabeling yield of the DMCA, which was found to be 96.1% for this specific radiolabeling experiment.

#### Hydrodynamic Size of [^99m^Tc]Tc-DPD-Fe_3_O_4_ DMCA

After radiosynthesis, the [^99m^Tc]Tc-DPD-Fe_3_O_4_ DMCA was evaluated for its hydrodynamic size. According to the obtained data presented in Figure 1c, two monomodal size distributions of [^99m^Tc]Tc-DPD-Fe_3_O_4_ DMCA with mean intensity weighted hydrodynamic diameters of about 300–450 nm were observed from two independent measurements. Considering that the DMCA exhibited a larger hydrodynamic diameter, compared to the one obtained from the parent, non-radiolabeled CA, DPD-Fe_3_O_4_ (D*_h_* = 96 nm) [28], we assume that this result is due to the radiolabeling conditions used [34].

### 3.2. In Vitro Stability Studies of [^99m^Tc]Tc-DPD-Fe_3_O_4_ DMCA

In vitro stability studies were also conducted on samples of [^99m^Tc]Tc-DPD-Fe_3_O_4_ DMCA, which had been previously incubated with PBS (pH 7.4) at room temperature and with human serum at T = 37 °C. In vitro stability was evaluated at four time points (1, 2, 4 and 24 h) after incubation by ITLC-SG, using acetone and sodium citrate solution (0.1 M) as the mobile phases. Quantitative in vitro stability data are presented in Figure 1d. Referring to the PBS solution, the DMCA exhibited high in vitro stability at all time points studied (94.3 ± 1.7%, 94.0 ± 1.4%, 94.1 ± 2.3% and 92.3 ± 4.0%, at 1, 2, 4, and 24 h of incubation). Referring to human serum, the DMCA exhibited satisfactory in vitro stability up to 4 h of incubation, while showing a slight decrease after 24 h of incubation (81.9 ± 2.5%, 76.1 ± 3.6%, 74.8 ± 3.2% and 67.3 ± 5.0%, at 1, 2, 4, and 24 h of incubation). In both cases, the time point 0 h corresponds to the initial radiolabeling yield of the samples, that is 95.8 ± 1.3% (see Table 2). The in vitro stability of [^99m^Tc]Tc-DPD-Fe_3_O_4_ DMCA evaluated in PBS (pH 7.4) confirmed the high stability of the DMCA even after an incubation period of 24 h. On the other hand, a slight decrease in stability of the DMCA was observed in human serum. Such a result is expected, since serum proteins may bind to ^99m^Tc, affecting the original DMCA. Thus, we may attribute the observed decreased stability in human serum to the presence of ^99m^Tc-binding or iron-binding (such as transferrin or ferritin) proteins. However, the serum stability study showed that no release of free [^99m^Tc]TcO_4_^−^ ions from the DMCA occurred even after 24 h, as demonstrated by the ITLC-SG analysis using acetone as the mobile phase.

### 3.3. In Vivo Biodistribution Study of [^99m^Tc]Tc-DPD-Fe_3_O_4_ DMCA in Normal Mice

A biodistribution study of the [^99m^Tc]Tc-DPD-Fe_3_O_4_ DMCA was conducted in normal mice to evaluate its in vivo biokinetics, as well as its main routes of clearance. For all biodistribution studies, high radiolabelling yields were achieved at 1 h post-incubation. Figure 2 presents quantitative data of the accumulation of the [^99m^Tc]Tc-DPD-Fe_3_O_4_ DMCA in the blood and organs of normal Swiss mice at 1, 2, 4 and 24 h p.i., expressed as the percentage of the injected activity per gram of tissue (% IA/g). According to the biodistribution data of the DMCA, the highest uptake appeared in the liver at all examined time points after intravenous injection (13.84 ± 5.64, 13.83 ± 4.65, 16.42 ± 3.27 and 10.18 ± 0.29% at 1, 2, 4 and 24 h p. i), while all other organs indicated lower uptake. The % IA/g of the DMCA in the kidneys was found to be 7.86 ± 1.64% up to 4 h p.i., while it decreased at 24 h p.i. (3.58 ± 0.22%). The% IA/g of the DMCA in the spleen remained practically stable at all time points (7.68 ± 1.86, 6.16 ± 3.84, 6.64 ± 2.78 and 5.88 ± 0.94 at 1, 2, 4 and 24 h p.i.). The blood retention of the DMCA was 3.45 ± 0.98% at 1 h p.i. and showed a decreasing pattern up to 24 h p.i. (0.59 ± 0.11%). Low accumulation of the DMCA in the heart and in the lungs was also observed, which both decreased over time following the blood pattern.

The results of the biodistribution of the [^99m^Tc]Tc-DPD-Fe_3_O_4_ DMCA in normal mice demonstrated noticeable uptake in the liver at all examined time points after intravenous injection, indicating the hepatobiliary system as a primary route of clearance. Additionally, a relatively significant level of radioactivity was found in the spleen and kidneys. The uptake in the main organs of the mononuclear phagocyte system (MPS), namely liver and spleen, is mainly attributed to the size of the DMCA. Specifically, the hydrodynamic diameter was found to be within the range of 300–450 nm, so the DMCA promptly undergoes opsonization after intravenous administration, which eventually results in phagocytic uptake by the MPS organs. The kidney uptake was not negligible up to 4 h p.i., indicating the renal route as a secondary way of excretion. Additionally, the low uptake observed in the stomach excludes the presence of [^99m^Tc]TcO_4_^−^ ions.

Several studies reported in the literature are in agreement with our biodistribution results [10,16,32,33,34,35,36]. For example, Tsiapa et al. [10] developed and assessed ^99m^Tc-labeled aminosilane-coated Fe_3_O_4_ nanoparticles as a dual-modality SPECT/MRI DMCA. The biodistribution in normal mice exhibited the greatest uptake in the liver and spleen at 1 and 24 h p.i. Specifically, the % IA/g in the liver and spleen was found to be 13.3 ± 1.9 and 9.4 ± 2.5 at 1 h p.i., respectively, while the % IA/g was found to be 9.8 ± 3.5 and 3.8 ± 2.7 at 24 h p.i. For this particular DMCA, the hepatobiliary system was found to be the main route of excretion, since the renal clearance was significantly low (2.7 ± 0.2% IA/g at 1 h p.i.). Lee et al. [16] examined the biodistribution of ^99m^Tc-labeled Fe_3_O_4_ nanoparticles by SPECT/CT at distinct time points in normal rats. The authors observed a high level of radioactivity in the liver shortly after the intravenous administration of the nanoparticles. The size of the radiolabeled nanoparticles was 275 nm, which is why they were mainly sequestered by the Kupffer cells of the liver. A significant uptake was also found in lungs, spleen, and bladder, while in the kidneys and heart, the uptake was minimal. Additionally, Mirković et al. [33] studied the biodistribution of ^99m^Tc-labeled bisphosphonate-coated Fe_3_O_4_ DMCA in normal mice and they showed significant liver and spleen uptake compared to the other organs. Specifically, the DMCA was rapidly cleared from the bloodstream and indicated high liver and spleen accumulation at all examined time points, namely 1, 2 and 24 h p.i. Indicatively, the highest uptake was observed at 1 h p.i. in the liver (18.0 ± 1.6% IA/g), followed by the spleen (12.0 ± 1.0% IA/g). This result is attributed to the fast opsonization that the DMCA undergoes, ultimately resulting in its phagocytosis by the macrophages. Moreover, Fu et al. [35] radiolabeled Fe_3_O_4_ nanoparticles with ^99m^Tc to serve as a potential DMCA in diagnostic applications. The DMCA was intravenously injected into normal rats and subsequently, its biodistribution was evaluated at 5, 30, 60 and 180 min p.i. The results indicated that almost 80–90% of the injected dose was taken up by the liver within 5 min after injection, while lower (compared to the liver) but significant uptake was observed in the lungs, kidneys and spleen. The high liver accumulation was associated with the phagocytosis that the DMCA undergoes by the macrophages, immediately after its administration to the blood circulation and was related to the physicochemical characteristics of the DMCA (i. e. size, surface charge, type and hydrophobicity of surface coating etc.). Shanehsazzadeh et al. [36] evaluated ^99m^Tc-labeled and dextran-coated iron oxide nanoparticles as a potential dual-modality SPECT/MRI contrast agent for breast cancer detection. According to the biodistribution results of the DMCA, the highest uptake appeared in the MPS organs, while all other organs indicated negligible uptake. The % IA/g in the liver was more than 60% at 15 min p.i. and decreased to approximately 24% at 24 h p.i. The spleen also exhibited the same biodistribution pattern, showing an uptake of more than 20% at 15 min p.i., which reduced to approximately 5% at 24 h p.i. The hydrodynamic size of the DMCA was found to be 115 nm by DLS and constitutes the main reason of opsonins absorption onto the surface of the DMCA, resulting in its accumulation in these organs.

### 3.4. In Vivo MR Imaging of [^99m^Tc]Tc-DPD-Fe_3_O_4_ in Normal Mice

#### 3.4.1. T_1_-Weighted In Vivo MRI Data

Referring to the in vivo MRI evaluation, Figure 3 illustrates representative T_1_-weighted (panels 3a,d) coronal and (panels 3e–h) axial MRI data of *n* = 4 normal Swiss mice focused on the area of interest (namely, liver and spleen, as indicated by the dotted red lines in all cases). All data were acquired 6 h p.i.

Figure 3a,e illustrate a reference mouse without any CA/DMCA, while Figure 3b,f show a mouse that was injected with the parent, non-radiolabeled DPD-Fe_3_O_4_ CA, at a concentration of C_CA_ = 0.1 mg/mL. A comparison of the MRI data (3b)/(3f) with the respective ones (3a)/(3e) reveals a negative contrast enhancement (loss of signal) in the area of interest (indicated by the dotted red lines), due to the accumulation of the CA in the region of MPS organs (i.e., liver and spleen).

Figure 3c,g illustrate a mouse that was injected with the [^99m^Tc]Tc-DPD-Fe_3_O_4_ DMCA at a concentration of C_DMCA_ = 0.01 mg/mL, while Figure 3d,h show a mouse that was injected with the DMCA at a ten times higher concentration of C_DMCA_ = 0.1 mg/mL. A comparison of the MRI data (3a)/(3e) with the respective ones (3c)/(3g) and (3d)/(3h) indicates that the [^99m^Tc]Tc-DPD-Fe_3_O_4_ DMCA displays darkening (or negative) contrast enhancement upon increase in its concentration. The comparable T_1_ MRI contrast effect exhibited at the same concentration C = 0.1 mg/mL by the [^99m^Tc]Tc-DPD-Fe_3_O_4_ DMCA (Figure 3d,h) and the parent DPD-Fe_3_O_4_ CA (Figure 3b,f) proves that the magnetic properties of the DMCA have not been altered by the radiolabeling process, thus preserving its contrast enhancement properties in vivo.

#### 3.4.2. T_2_-Weighted In Vivo MRI Data

Figure 4 illustrates the representative T_2_-weighted (panels 3a,d) axial MRI data of *n* = 4 normal Swiss mice focused on the area of interest, the liver and spleen (red dotted lines). All data were acquired 6 h p.i.

Panel 3a illustrates the area of interest of a reference mouse without any CA/DMCA, while panel 3b shows the area of interest of a mouse that was injected with the parent, non-radiolabeled DPD-Fe_3_O_4_ CA, at a concentration of C_CA_ = 0.1 mg/mL. A comparison of these two panels reveals a negative contrast enhancement, due to the accumulation of the CA in this organ.

Panel 3c illustrates the area of interest of a mouse injected with the [^99m^Tc]Tc-DPD-Fe_3_O_4_ DMCA at a concentration of C_DMCA_ = 0.01 mg/mL, while panel 3d shows the area of interest of a mouse injected with the DMCA at a ten times higher concentration of C_DMCA_ = 0.1 mg/mL. Comparing the MRI data 3c and 3d with the respective 3a, we can clearly observe that the DMCA provides a negative contrast enhancement that is concentration dependent. Additionally, by comparing the negative contrast effect exerted by the [^99m^Tc]Tc-DPD-Fe_3_O_4_ DMCA at the concentration of C_DMCA_ = 0.1 mg/mL (panel 3c) with the respective one caused by the non-radiolabeled DPD-Fe_3_O_4_ CA (panel 3b), we can conclude that the magnetic properties of the parent CA, DPD-Fe_3_O_4_, are not suppressed by the radiolabeling process.

### 3.5. In Vivo Gamma-Camera Imaging of [^99m^Tc]Tc-DPD-Fe_3_O_4_ in Normal Mice

Gamma-camera imaging was conducted in normal Swiss mice to further evaluate the in vivo efficiency, as well as the biokinetics of the [^99m^Tc]Tc-DPD-Fe_3_O_4_ DMCA. The mice were injected with 100 μL/0.02 mg/7.03 MBq of DMCA, leading to a final concentration of C_DMCA_ = 0.01mg/mL. Figure 5 illustrates the representative coronal gamma-camera images obtained at 1 h (panel 5a), 2 h (panel 5b) and 4 h (panel 5c) p.i. A simple comparison of the imaging data indicates that the [^99m^Tc]Tc-DPD-Fe_3_O_4_ DMCA mostly accumulated in the region of the liver and spleen at all examined time points; hence, this is in good agreement with the respective biodistribution and MRI results.

## 4. Conclusions

In this study, we investigated the radiosynthesis and subsequent in vitro and in vivo evaluation of [^99m^Tc]Tc-DPD-Fe_3_O_4_ as a potential SPECT/MRI DMCA. DPD-Fe_3_O_4_ was radiolabeled with ^99m^Tc via the direct radiolabeling method, resulting in high radiolabeling yields (~96%). [^99m^Tc]Tc-DPD-Fe_3_O_4_ remained stable up to 24 h post-preparation, as determined by the in vitro stability study after incubation of the DMCA with PBS at room temperature and human serum at 37 °C, showing 92.3% and 67.3% intact radiolabeled DMCA, respectively. The subsequent ex vivo biodistribution studies in normal mice presented noticeable liver uptake, followed by uptake in the kidneys and spleen at lower percentages. Both gamma camera and MR imaging studies performed in normal mice confirmed the highest accumulation of the DMCA in the area of the liver, in agreement with the obtained biodistribution results. In conclusion, our preliminary in vitro and in vivo findings clearly demonstrate the potential of [^99m^Tc]Tc-DPD-Fe_3_O_4_ as a SPECT/MRI DMCA.

## Figures and Tables

**Figure 1 nanomaterials-12-02728-f001:**
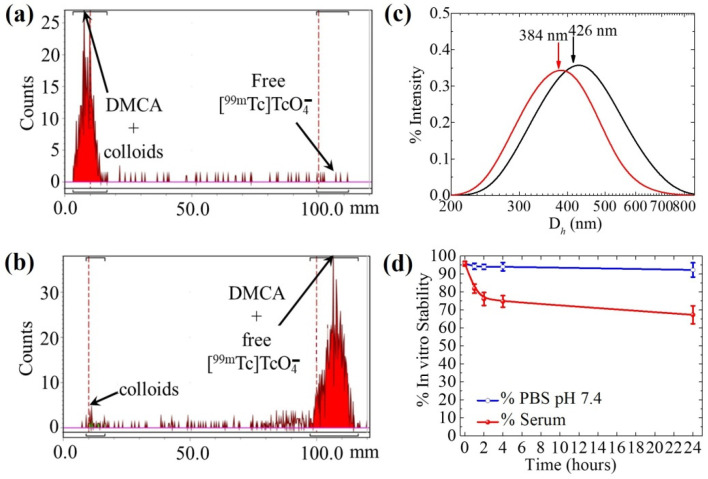
Radio-chromatograms of [^99m^Tc]Tc-DPD-Fe_3_O_4_ DMCA applied on ITLC-SG strips and developed with (**a**) acetone and (**b**) sodium citrate solution (0.1 M); (**c**) hydrodynamic diameter (D*_h_*) of [^99m^Tc]Tc-DPD-Fe_3_O_4_ DMCA, as determined by DLS. Two independent measurements are illustrated; (**d**) in vitro stability of [^99m^Tc]Tc-DPD-Fe_3_O_4_ DMCA incubated in PBS (pH 7.4) on a shaker at RT and in human serum at T = 37 °C, as both determined by ITLC-SG chromatographic analysis at 1, 2, 4 and 24 h of incubation. The percentage (%) of in vitro stability corresponds to the overall mean value ± standard deviation (MV ± SD) of three independent experiments.

**Figure 2 nanomaterials-12-02728-f002:**
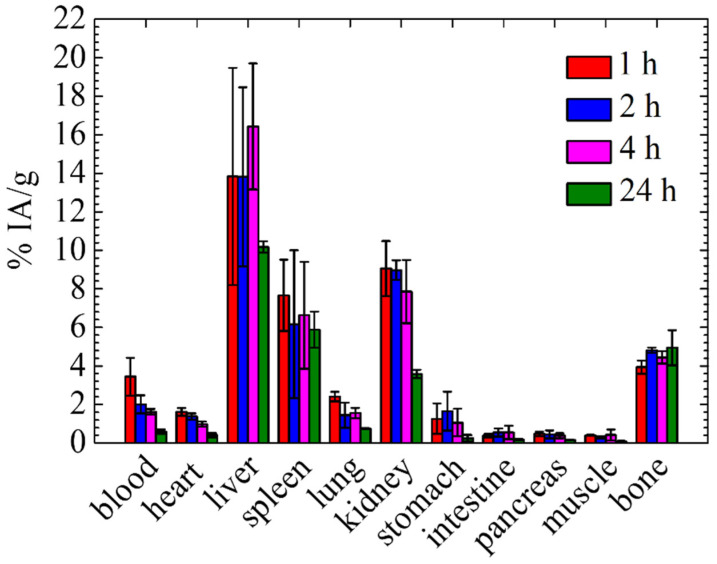
The accumulation of the [^99m^Tc]Tc-DPD-Fe_3_O_4_ DMCA in the organs of normal Swiss mice, expressed as percentage injected activity per gram of tissue (% IA/g) at 1, 2, 4 and 24 h post injection (p.i.). For each time point, three mice were studied; thus, the results are expressed as the MV ± SD.

**Figure 3 nanomaterials-12-02728-f003:**
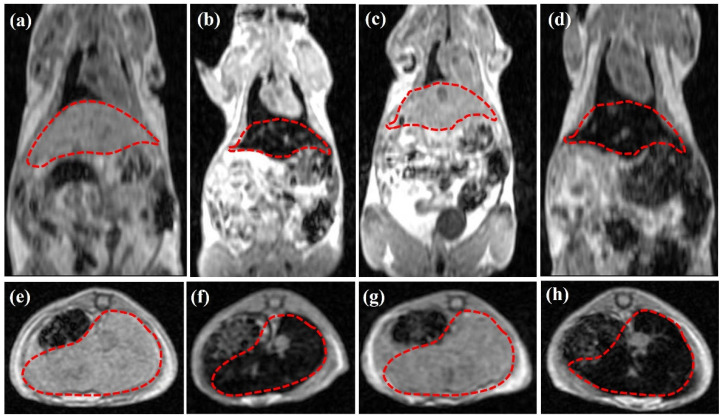
Representative T_1_-weighted (**a**–**d**) coronal and (**e**,**f**) axial MRI data of *n* = 4 normal Swiss mice: (**a**,**e**) without any CA/DMCA; (**b**,**f**) with non-radiolabeled DPD-Fe_3_O_4_ CA at a concentration of C_CA_ = 0.1 mg/mL; (**c**,**g**) with [^99m^Tc]Tc-DPD-Fe_3_O_4_ DMCA at C_DMCA_ = 0.01 mg/mL; (**d**,**h**) with [^99m^Tc]Tc-DPD-Fe_3_O_4_ DMCA at C_DMCA_ = 0.1 mg/mL. All MRI images were acquired 6 h p.i. The dotted red lines indicate the area of interest (liver and spleen).

**Figure 4 nanomaterials-12-02728-f004:**
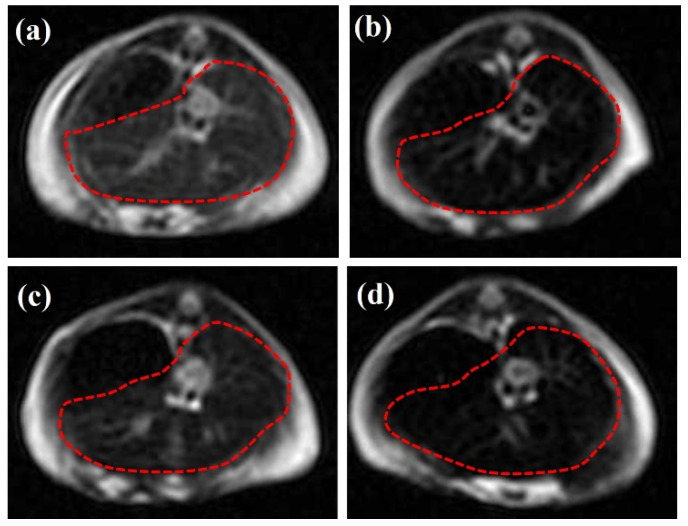
Representative T_2_-weighted (**a**–**d**) axial MRI data of the liver of *n* = 4 normal Swiss mice: (**a**) without any CA/DMCA; (**b**) with non-radiolabeled DPD-Fe_3_O_4_ CA at a concentration of C_CA_ = 0.1 mg/mL; (**c**) with [^99m^Tc]Tc-DPD-Fe_3_O_4_ DMCA at C_DMCA_ = 0.01 mg/mL; (**d**) with [^99m^Tc]Tc-DPD-Fe_3_O_4_ DMCA at C_DMCA_ = 0.1 mg/mL. The dotted red lines indicate the area of interest (liver and spleen).

**Figure 5 nanomaterials-12-02728-f005:**
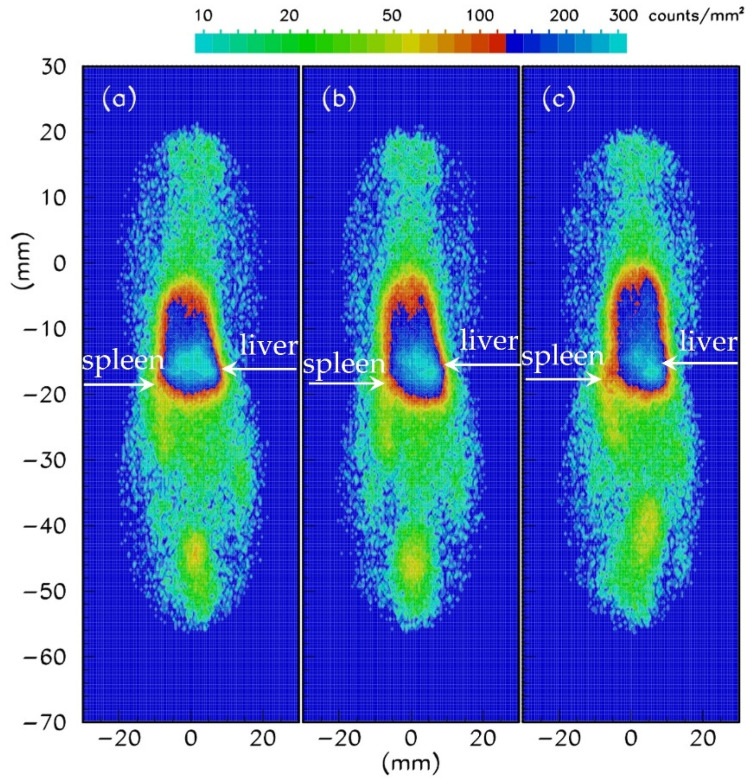
(**a**–**c**) Representative coronal gamma camera images of a normal Swiss mice intravenously injected with [^99m^Tc]Tc-DPD-Fe_3_O_4_ DMCA at a final concentration of C_DMCA_ = 0.01 mg/mL. The images were obtained at (**a**) 1, (**b**) 2 and (**c**) 4 h p.i.

**Table 1 nanomaterials-12-02728-t001:** Percentage (%) of radiolabeling yield of [^99m^Tc]Tc-DPD-Fe_3_O_4_ DMCAs incubated with different pH values, as a function of incubation time at room temperature (RT) and T = 50 °C.

pH	Temperature (°C)	30 (min)	60 (min)	120 (min)
2	RT 50 °C	87 60	90 23	94 ^1^ 53
4	RT 50 °C	80.2 60	85.2 67	88 70
7	RT	96.3 ^1^	96.4 ^1^	97.2 ^1^

^1^ Radiolabeling yields of high value, close to the ideal 100%, have been carefully checked for reproducibility.

**Table 2 nanomaterials-12-02728-t002:** Percentage (%) of radiolabeling yield of [^99m^Tc]Tc-DPD-Fe_3_O_4_ DMCAs incubated at pH 2 and 7, as a function of incubation time at room temperature (RT). The results are expressed as mean value ± standard deviation of at least three independent experiments.

pH	Temperature (°C)	30 (min)	60 (min)	120 (min)
2	RT			89.5 ± 6.9
7	RT	95.9 ± 0.8	95.8 ± 1.3	96.1 ± 1.8

**Table 3 nanomaterials-12-02728-t003:** Percentage (%) of in vitro bench stability as a function of time of [^99m^Tc]Tc-DPD-Fe_3_O_4_ DMCAs initially prepared at two different pH values, namely at pH 2 and at pH 7, respectively. The results are expressed as mean value ± standard deviation of at least three independent experiments.

pH	0 (min)	60 (min)	120 (min)	240 (min)
2	89.5 ± 6.9	74.3 ± 14.2	69.9 ± 18.9	71.3 ± 21.9
7	95.8 ± 1.3	96.3 ± 0.6	96.1 ± 1.8	94.0 ± 1.7

**Table 4 nanomaterials-12-02728-t004:** Effect of the amount (mg) of SnCl_2_ on the radiolabeling yield (%) [^99m^Tc]Tc-DPD-Fe_3_O_4_ DMCA, incubated at pH 7 for 60 min at RT. The respective % of free [^99m^Tc]TcO_4_^−^ and colloidal ^99m^Tc are also indicated. The results are expressed as mean value ± standard deviation of independent experiments.

SnCl_2_ (mg)	[^99m^Tc]Tc-DPD-Fe_3_O_4_ (%)	[^99m^Tc]TcO_4_^−^ (%)	Colloids (%)
8.7 ± 0.6	96.2 ± 0.9	0.5 ± 0.3	3.3 ± 0.7

## Data Availability

The data presented in this study are available upon request from the corresponding author D.S.

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
