# Peer review of "99mTc-Labeled Iron Oxide Nanoparticles as Dual-Modality Contrast Agent: A Preliminary Study from Synthesis to Magnetic Resonance and Gamma-Camera Imaging in Mice Models"

_nanomaterials, 2022, doi:10.3390/nano12152728_

Round 1

Reviewer 1 Report

The manuscript submitted for review presents the continuation of experimental work initiated by this group of authors in order to develop dual-modality contrast agent based on functionalized iron oxide nanoparticles for MRI and single photon emission tomography, SPECT.  It can be accepted for publication in Nanomaterials after a minor review. In particular the authors are requested to address the following issues:

Title: this work presents the very preliminary results of the technetium-99m labelled iron oxide nanoparticles functionalized with 2,3-dicarboxypropane-1,1-diphosphonic acid. This preliminary character of the study should be reflected in the title of manuscript.

Introduction:

Page 2, lines 57-58 – the sentence refers to the use of DMCAs in hyperthermia but there is no reference provided to support that statement.  

Page 2, lines 59-65 – this paragraph presents the role of technetium-99m radiopharmaceuticals in diagnostic imaging. However, without mentioning the (99mTc) 3,3-diphosphono-1,2-propanedicarboxylic acid (DPD) injection (TECEOS®), a diphosphonate which was initially developed as bone imaging agent and is currently explored in diagnosis of cardiac amyloidosis.  Some more information about this radiopharmaceutical as well as about the agent used to functionalize surface of Fe3O4 nanoparticles would help the reader to understand the differences/similarities between these two tracers.

Results and discussion:

Page 5, Table 1. The use of elevated temperature to increase the radiolabeling yield did not result in this expected result – in contrary – the yields were much lower. This observation is not typical for the diphosphonates radiolabeled with technetium-99m. What is the explanation of such behavior?

Page 5, Table 1. In the text the authors inform, that the highest radiolabeling yields are indicated by black asterisk but there is no black asterisk in the Table. Also, the text in the footnote to this Table is not clear and confusing – please make it more specific.

Page 6, Tables 2 and 3 – the presentation of results without standard deviations suggests that it was only single result per point. Please explain.

Page 9, line 449, the wording used “For the specific DMCA..” is unfortunate – what do the authors mean by “specific  DMCA”? The lines 444-451 need a careful check for clarity.

Author Response

Please see the uploaded file, Reply to Reviewer 1.

Reviewer 2 Report

The authors presented a study on their preparation of a dual mode contrast agent, Fe3O4-Tc, for SPECT/MRI, which seems to show good stability in vitro and good imaging in vivo. The main question is what makes their nanoparticle different from other similarly reported particles? 

1. The authors need to mention what makes their work unique, in particular their DMCA Fe3O4-Tc, compared to previous reports in literature since these were already done before.

Is there any advantage/disadvantage to their formulation? What makes their DMCA nanoparticle special?

2. In methods (section 2.1), does the concentration of Fe3O4 refer to mg Fe or mg Fe3O4 per mL? The same for section 2.4, the concentration in mg does not refer to anything specific?

Is this in terms of Fe or the weight of the material? And if this in terms of the total weight of the nanoparticle, how were these concentration accurately measured?

3. Given that these are nanoparticles, it would be really helpful if the authors can provide TEM images of the nanoparticles and compared thesize obtained from TEM with DLS.

4. Section 3.1, asterisks were mentioned in the text to indicate highest radiolabelling but the superscript "1" instead was used in Table 1. 

5. I don't understand why pH = 2 was used to evaluate stability when acidic pH is known to oxidize Fe3O4 particles.

a. The authors mention that dissociation but could this be simply due to Fe3O4 being completely oxidized in highly acidic conditions? 

b. Otherwise, could the authors show DLS or TEM images to show the stability of particles in pH = 2 conditions?

6. It is rare that hydrodynamic radius is reported for the size of the  nanoparticles instead of hydrodynamic diameter, which is a better way to describe particle size instead. If possible, please change.

7. In Figure 2C, no y axis label. In Figure 2D, no value for time = 0 h, which is the baseline for comparison I believe with the other timepoints? Could the authors provide the exact value as well and not just percentage? The error bars are barely visible as well.

8. In Figure 3, please add a scale bar. In addition, could the authors draw a more previse area of interest? The blue circle drawn is very broad.

9. In Figure 4, please indicate with arrows or draw an ROI of the location of the liver in the image. Please add a scale bar.

10. In Figure 5, there is no scale for the signal intensity of the images. In addition, no scale bar with respect to the size of the animal.

Also, please point with an arrow or draw an ROI to label the organs of particular interest especially the liver and spleen where majority of the particles accumulated.

Author Response

Please, see the uploaded file, Reply to Reviewer 2.

Round 2

Reviewer 2 Report

Thank you to the authors for addressing my comment. With regard to measurement of concentration of the particle, if possible, I would recommend that the authors use more accurate common techniques such as atomic absoprtion spectroscopy (AAS) or ICP-MS, to measure the concentration of their particles in vitro and in vivo (in mouse blood) given that there is Fe in their nanoparticle, which can be measured with the aforementioned techniques. Gravimetric analysis of sample weight is very imprecise given the low mass being measured (mg scale). 

Author Response

Please, see the uploaded file.
